# Oral health and oral health-related quality of life in patients with chronic peripheral facial nerve palsy with synkineses—A case-control-study

Lisa Strobelt[1,2,3]*, Anna-Maria Kuttenreich[2,3,4], Gerd Fabian Volk[2,3,4], Carien Beurskens[5], Thomas Lehmann[6], Ina Manuela Schüler[1]

1 Section for Preventive Dentistry and Pediatric Dentistry at the Center for Dentistry, Oral and Maxillofacial Surgery of Jena University Hospital, Jena, Germany, 2 Department of Otorhinolaryngology, Jena University Hospital, Jena, Germany, 3 Facial-Nerve-Center Jena, Jena University Hospital, Jena, Germany, 4 Center for Rare Diseases Jena, Jena University Hospital, Jena, Germany, 5 Department of Physiotherapy, Radboud University Medical Center, Nijmegen, The Netherlands, 6 Institute of Medical Statistics, Computer and Data Sciences, University Hospital Jena, Jena, Germany

* lisa.strobelt@gmx.de

**Data Availability Statement:** All relevant data are within the paper and its Supporting Information files.

## Abstract

Facial nerve palsy (FP) is the most common cranial nerve lesion, leading to partial or complete immobility of the affected half of face. If food residues on tooth surfaces cannot be removed by natural self-cleaning mechanisms that this is likely to lead to an increase dental plaque formation and the risk of dental, periodontal and general diseases. It was the aim of this study to assess oral health, oral hygiene with the influence of handedness and oral health related quality of life in patients with chronic peripheral FP. This study included 86 people. Patients with FP (n = 43) in an FP-group (FPG) were matched with 43 participants without ever diagnosed FP in a control group (CG). Oral health and oral hygiene were assessed in a clinical examination performed in hospital. Decayed-missing-filled-teeth-index, periodontal-screening-index, approximal plaque index and sulcus bleeding index were used to evaluate dental caries and periodontal health status. Oral health-related quality of life data (OHQoL) was collected with the Oral health impact profile (Germany-14) by interview. Oral health parameters in the FPG were significantly worse than in the CG. The median revealed 14.3% more proximal plaque (p = 0.014), 20.8% more sulcus bleeding (p = 0.002) and more than twice as much caries (p = 0.024). The paretic side compared to the non-paretic side of patients was significantly more affected by inflammatory periodontal diseases (p = 0.032) and had a higher prevalence of caries (p = 0.163). Right-handed patients with right-sided FP performed worse than right-handed patients with left-sided FP (p = 0.004). Patients with FP described a greater limitation of OHQoL than patients without this disease (p < 0.001). In conclusion, facial nerve palsy affects oral health, oral hygiene and OHQoL of patients while handedness influences oral hygiene and oral health.

**Funding:** The author(s) received no specific funding for this work.

**Competing interests:** The authors have declared that no competing interests exist.

## Introduction

Facial nerve palsy (FP) is the most common cranial nerve lesion [1] with an assumed incidence of 30/ 100.000 [2]. The main symptom of acute paresis is flaccid one-sided immobility of facial muscles [3]. In at least 30% of acute cases, there is no complete recovery. Instead incorrect reinnervation with increased and miscoordinated muscle activity has been observed [4]. It is often the perioral region that is also affected, not only in case of the infranuclear damage examined in this study, the so-called peripheral facial paresis, but also in the much more common central paresis [5]. The muscle tension in the buccinator muscle is missing, is too high or uncoordinated after incorrect reinnervation, which can result in bite injuries in patient's cheek area. In addition, there is a risk that if there is no or incorrect activity in the cheek after eating, food particles remain in the cheek pouch, which promotes inflammation of the mucous membrane [5]. In the area of the lip muscles, the flaccid paralysis of facial muscles results in the corner of the mouth drooping on the affected side and thus preventing complete lip closure. For example, patients cannot hold the liquid in their mouths when drinking [3].

A previous study [6], including a quantitative evaluation of oral function in the acute and recovery phase of idiopathic facial palsy, has provided an initial indication that peripheral facial paralysis impairs oral hygiene and can promote oral disease. However, even after reinnervation, the coordination of the lips and the corner of the mouth is a long-term problem for most patients.

Due to the lack of evidence, the question remains as to what exactly oral hygiene looks like in patients with FP, whether deficits in oral health become apparent due to the more difficult conditions of oral care, whether these correlate with the duration of the disease, and whether differences in oral health between the paretic (PS) and non-paretic side (NPS) can be recorded. Furthermore, it has not been clarified yet whether handedness of affected patients has an influence in this regard. Patients with FP should ideally receive a wide range of therapeutic measures: speech therapy, physiotherapy, occupational therapy and psychological assistance. In addition great importance is placed on eye protection [7]. In contrast, oral health has not been considered yet as part of preventive and therapeutic care strategy.

This study aims to compare oral health between patients with and without FP. The endpoints to be analysed were, as primary criterion, the evaluation of periodontal health using the maximum periodontal screening index (PSI), as well as further various oral health parameters (see 2.3) and the oral health related quality of live (OHQoL) as a secondary outcome.

The following hypotheses were tested:

1. **H0:** Oral health and oral health related quality of life does not differ between patient with and without FP.

   **Ha:** Oral health and oral health related quality of life differs between patient with and without FP.

2. **H0:** In patients with FP, oral disease does not affect the PS more than the NPS.

   **Ha:** In patients with FP, oral disease affects the PS more than the NPS.

3. **H0:** Oral health is not influenced by the individual oral hygiene behaviour of patients with FP.

   **Ha:** Oral health is influenced by the individual oral hygiene behaviour of patients with FP.

4. **H0:** The oral health and oral hygiene behaviour of patients with FP do not depends on patient-related factors, such as the handedness, the duration and perception of the disease and the disease-related quality of life.

**Ha:** The oral health and oral hygiene behaviour of patients with FP depend on patient-related factors, such as the handedness, the duration and perception of the disease and the disease-related quality of life.

The present study contributes to the evidence regarding oral health and oral hygiene of patients with FP in a controlled design, analysing influencing factors and deriving recommendations for dental and general medical care.

## Materials and methods

### Recruitment and inclusion and exclusion criteria

Consecutive patients with postparalytic synkineses after acute unilateral peripheral FP were recruited and consecutively included from 01/09/2020 to 14/10/2020 in the Facial-Nerve-Center of Jena University Hospital. This study was approved by Jena University Hospital Ethics Committee (Reg. 2020-1883-BO) on 27/08/2020. To detect an effect size of Cohen's d = 0.67 with a power of 80%, a total of 86 patients (43 per group; Fig 1) need to be analysed (G*Power 3.1, two sided Wilcoxon-Mann-Whitney Test, significance level 5%). Each patient with FP was matched by age and gender to one control. With regards to the match parameter age, an age range of 5 years was applied. All consecutive included patients and healthy controls gave their informed consent and carried out the study to the end. Edentulous, multimorbid patients, as well as those with diabetes mellitus type 1 and 2, hemophilia A and B, epilepsy and AIDS, were excluded from the study. An existing or a past FP was considered as further exclusion criteria for the CG. CG-patients were selected if matched to FPG-patients by age and gender.

### Calibration of the examiner

Before the clinical study, theoretical and practical calibration training of the investigating dentist (L. S.) under the supervision of a clinically and epidemiologically experienced dentist (I. M. S.) was carried out for the Decayed-missing-filled-teeth-index (DMFT) and Turesky-Index using photographs. To compare the two examiners, the kappa statistic [8] was applied and the calculated value was evaluated using Landis/Koch [9].

### Oral examination

The oral examination took place in a treatment room of the Facial-Nerve-Center at agreed examination appointments as part of the daily clinic routine, without the patients performing oral hygiene beforehand. The patients sat upright on a conventional chair. The examiner (L. S.), a calibrated dentist, used a blunt probe (periodontal probe; piece 973/CP15; Manufacturer Article no: 973/CP15; Item no.: 19329; Carl Martin GmbH; Solingen; Germany) and a mirror mouth mirror standard no. 4 hollow; Manufacturer Article no. 70878; Item no. 70878; OMNI-DENT Dental-Handelsgesellschaft mbH; Rodgau/Nieder-Roden; Germany. Each examination was performed by the same examiner. The oral cavity was illuminated with a head lamp (Faro-MED-Effner; 08-365-GR; 4000–6500 LUX; Berlin; Germany). The following oral health parameter were collected and documented: tooth mobility (TM) [10], periodontal screening index (PSI) [5], maximum vestibular probing depth (VPD in mm), attachment loss (AL in mm), modified sulcus bleeding index (SBI) [11], papillary bleeding index (PBI) [12], approximal plaque index (API) [13], Turesky plaque index (TI) [14], presence of tartar (POT) and dental caries by DMFT-index [15]. MIRA2Ton staining solution (Hager & Werken GmbH & Co. KG; Duisburg; Germany) was used to stain the plaque.

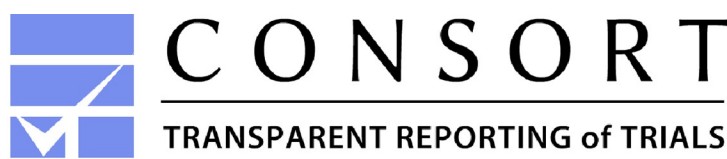

**Enrollment**

Assessed for eligibility (n = 114)

Excluded (n = 28)
➢ Not meeting inclusion criteria (n = 0)
➢ Declined to participate (n = 28)
➢ Other reasons (n = 0)

Study population (n = 86)

**Allocation**

The participating persons did not receive any intervention. A single oral examination was carried out in 2 different groups:
**Group 1**: patients with peripheral facial nerve palsy
Allocated to oral examination (n = 43)
➢ Received allocated oral examination (n = 43)
➢ Did not receive allocated oral examination (n = 0)

The participating persons did not receive any intervention. A single oral examination was carried out in 2 different groups:
**Group 2:** controls
Allocated to oral examination (n = 43)
➢ Received allocated oral examination (n = 43)
➢ Did not receive allocated oral examination (n = 0)

**Analysis**

➢ Analysed (n = 43)
➢ Excluded from analysis (n = 0)

➢ Analysed (n = 43)
➢ Excluded from analysis (n = 0)

**Fig 1. CONSORT flowchart.**

### Surveys

Data regarding individual oral hygiene behaviour and OHQoL were collected by questionnaire. Patients were asked about their oral hygiene behaviour and their attitudes towards oral hygiene using a specially developed questionnaire. The validated OHIP-G14 [16] questionnaire served to record the OHQoL. The handedness was determined with the Edinburgh Handiness Inventory (EHI) [17] questionnaire. All questionnaires were answered by interview.

## Medical data collection

The values of the Facial Disability Index (FDI) [18, 19] and the Facial Clinimetric Evaluation Scale (FaCE) [19, 20], evaluating the disease related quality of life, were taken from medical records. The FDI includes a physical function scale, ranging from 25 (most severe impairment) to 100 (no impairment) and a social function scale with a minimum of 0 (most severe social impairment) and a maximum of 100 (no social impairment). The FaCE scale covers a range from 0 (most severe impairment) to 100 (no impairment).

## Data collection and statistical analysis

A pseudonymised data collection was carried out with the help of a patients/controls identification list. Each patient was assigned an identification number. This list was stored in accordance with data protection requirements and all data analysis was only carried out with the pseudonymised data records. The paper-based examination sheets and questionnaires were digitalised and data recorded in Microsoft Excel (Version 2007) by the investigating dentist. Only the participating authors had access to the collected data. To test the normal distribution, we used the Shapiro-Wilk-test. To check the significance, the Mann-Whitney-U-test was used for differences in the group comparisons between the case group and the control group, and the Wilcoxon-test for differences between the paretic and non-paretic side of each individual. In order to reduce the problem of multiple testing, we corrected our values with the Bonferroni correction. A linear multivariable regression analysis tested the influence of selected factors (duration of the illness, disease-related quality of life, perception of the patient in relation to the OHQoL) on individual oral health parameters. Mutual relationships between individual oral health parameters and answers to the survey were determined using Spearman's bivariate correlation analysis. The level of significance was set at $\alpha = 0.05$. All statistical analysis was carried out using IBM SPSS Statistics (IBM Released 2020. IBM SPSS Statistics for Windows, Version 27.0. Armonk, NY) and evaluated by the investigating dentist (L.S.) and a statistician (T. L.).

## Results

### Study population

A total of 86 people aged between 21 and 77 years, including 58 females (67.4%), were examined (Table 1). The median age of both groups was 51.0 years (IQR 20.75). The age in both groups was normally distributed according to the Shapiro-Wilk-Test. Patients with right-sided FP were on average 11.6 years older than patients with left-sided FP (p = 0.05). This significant difference existed in both males (16.6 years) and females (8.3 years). With regard to

**Table 1. Age and gender of the patients with peripheral facial nerve palsy and the control group.**

| Criteria | | FPG | CG | Total |
|---|---|---|---|---|
| **N (%)** | total | 43 (100.0) | 43 (100.0) | 86 (100.0) |
| **Gender** N (%) | male | 14 (32.6) | 14 (32.6) | 28 (100.0) |
| | female | 29 (67.4) | 29 (67.4) | 58 (100.0) |
| **Age** (in years) Median(IQR) [Min; Max] | total | 51.0 (4.0) [21;73] | 51.0 (5.5) [23;77] | 51.0 (20.75) [21;77] |
| | male | 50.0 (6.0) [24;73] | 48.5 (18.0) [24;77] | 49.3 (29.50) [24;77] |
| | female | 51.0 (5.5) [21;68] | 52.0 (4.0) [23;71] | 51.0 (16.25) [21;71] |

Age and gender of the patients with peripheral facial nerve palsy (FPG) and the control group (CG), given in number and percent or median with interquartile range as well as minimum and maximum value

**Table 2. Comparison of oral health parameters between patients with peripheral nerve palsy and the control group in regard to the primary endpoint.**

| Oral health parameters | | FPG | CG | corrected p-value |
|---|---|---|---|---|
| Periodontal health | VPD (mm) Median (IQR) [Min; Max] | **2.3** * (0.9) [0.6; 3.2] | 2.0 (0.8) [1.0; 3.3] | **0.428** |
| | AL (mm) Median (IQR) [Min; Max] | **2.2** (0.9) [0.7; 4.6] | 2.1 (1.0) [1.2; 5.5] | 1.000 |
| | PSI MAX Median (IQR) [Min; Max] | **2.2** (0.8) [0.6; 3.2] | 2.0 (0,7) [1.2; 3.0] | 1.000 |
| | TM $\bar{x}$ (SD) [Min; Max] | 0.03 (0.13) [0.0; 0.8] | 0.03 (0.09) [0.0; 0.4] | 1.000 |

*p≤0,05 for side comparison between patients with peripheral facial nerve palsy (FPG) and the control group (CG) using the Mann-Whitney U-Test regarding the oral health parameters vestibular probing depth (VPD), attachment loss (AL), periodontal screening index (PSI MAX), tooth mobility (TM), information by means of median and interquartile range (IQR) or mean value ($\bar{x}$) and standard deviation (SD) as well as minimum and maximum [Min; Max] and bilateral asymptotic corrected significance p (Bonferroni correction).

handedness, 93.0% (n = 40) of the FPG and 90.7% (n = 39) of the CG were right-handed. The duration of disease ranged from 6 months to 64.8 years (congenital FP) with a median of 24.0 months. Patients in the FPG recorded a subjectively perceived disease related quality of life of 69.0 (median FDI) and 60.0 (median FaCE). Causative factors for unilateral FP with miscoordinated muscle activity were surgically removed benign tumors (vestibular schwannomae) in 37.2% (n = 16) of the patients, viral diseases in 18.6% (n = 8) of the patients and trauma in 4.7% (n = 2) of the patients. FP occurred idiopathically in 15 (34.9%) and congenitally in two (4.7%) of the patients.

## Oral findings

**Comparison of oral health between patients with peripheral facial nerve palsy and the control group.** No significant differences of periodontal health were found between FPG and CG (Table 2). VPD, AL, PSI, and the number of missing teeth tended to have poorer values in FPG-patients, but without statistical significance (Table 2).

However, between FPG and CG, significant differences in plaque contamination, bleeding and prevalence of untreated dental caries were determined (Table 3). Plaque contamination was significantly higher in the FPG than in the CG (Table 3). Bleeding as a clinical sign of gingival inflammation was also observed significantly more frequently in the FPG than in the CG

**Table 3. Comparison of oral health parameters between patients with peripheral nerve palsy and the control group in regard to the secondary endpoints.**

| Oral health parameters | | FPG | CG | Corrected p-value |
|---|---|---|---|---|
| Calculus | Median N (Teeth) (IQR) [Min; Max] | 4.0 (4.0) [0.0; 10.0] | 4.0 (3.0) [0.0; 16.0] | 0.965 |
| Plaque | API (%) Median (IQR) [Min; Max] | **53.6*** (23.7) [14.3; 100.0] | 39.3 (25.0) [7.1; 75.0] | **0.014** |
| | TI $\bar{x}$ (IQR) [Min; Max] | **0.6*** (0.5) [0.0; 1.0] | 0.4 (0.3) [0.0; 1.3] | **0.004** |
| Gingival inflammation | SBI (%) Median (IQR) [Min; Max] | **58.3*** (35.7) [10.7; 100.0] | 37.5 (25.9) [3.6; 85.7] | **≤ 0.002** |
| | PBI (%) Median (IQR) [Min; Max] | **26.0*** (26,0) [3.0; 62.0] | 16.0 (13,0) [0.0; 45.0] | **≤ 0.002** |
| Untreated caries | DT $\bar{x}$ (SD) [Min; Max] | **1.3*** (2.0) [0.0; 7.0] | 0.5 (0.0) [0.0; 5.0] | **0.024** |
| Missing teeth | MT $\bar{x}$ (SD) [Min; Max] | **2.3** (3.7) [0.0; 20.0] | 1.9 (3.4) [0.0; 19.0] | 0.663 |
| Restored teeth | FT Median (IQR) [Min; Max] | 10.0 (8.0) [0.0; 20.0] | 11.0 (7.0) [0.0; 21.0] | 0.459 |

*p ≤ 0,05 for side comparison between patients with peripheral facial nerve palsy (FPG) and the control group (CG) using the Mann-Whitney U-Test regarding the oral health parameters, number of teeth with calculus, approximal plaque index (API), Turesky index (TI), sulcus bleeding index (SBI), papillary bleeding index (PBI), number of untreated caries (decayed teeth; DT), number of missing teeth (missed teeth; MT) as well as number of restored teeth (filled teeth; FT), information by means of median and interquartile range (IQR) or mean value ($\bar{x}$) and standard deviation (SD) as well as minimum and maximum [Min; Max] and bilateral asymptotic corrected significance p (Bonferroni correction).

**Table 4. Side comparison between the paretic and non-paretic side of the patients with peripheral nerve palsy.**

| Oral health parameters | | PS | NPS | Corrected p-Value |
|---|---|---|---|---|
| Calculus | Median N(Teeth) (IQR) [Min; Max] | 2.0 (2.0) | 2.0 (2.0) | 0.248 |
| | | [0.0; 5.0] | [0.0; 5.0] | |
| Plaque | API (%) Median (IQR) [Min; Max] | **57.1*** (26.0) | 46.2 (33.0) | **0.004** |
| | | [14.3; 100.0] | [14.3; 100.0] | |
| | TI $\bar{x}$ (IQR) [Min; Max] | **0.6*** (0.6) | 0.4 (0.5) | **≤0.002** |
| | | [0.0; 1.9] | [0.0; 1.5] | |
| Gingival inflammation | SBI (%) Median (IQR) [Min; Max] | **64.3*** (35.7) | 57.1 (41.2) | 0.068 |
| | | [7.1; 100.0] | [14.3; 100.0] | |
| | PBI (%) Median (IQR) [Min; Max] | **15.0*** (13.0) | 11.0 (16.0) | **≤0.002** |
| | | [2.0; 35.0] | [1.0; 30.0] | |
| Periodontal health | VPD (mm) Median (IQR) [Min; Max] | 2.4 (1.0) | 2.3 (0.8) | 0.088 |
| | | [0.7; 3.3] | [0.6; 3.4] | |
| | AL (mm) Median (IQR) [Min; Max] | 2.5 (1.0) | 2.4 (1.0) | 0.0564 |
| | | [0.7; 4.3] | [0.7; 4.9] | |
| | PSI $\bar{x}$ (SD) [Min; Max] | **2.4*** (0.4) | 2.2 (0.5) | **0.0324** |
| | | [1.5; 3.0] | [1.0; 3.0] | |
| | PSI Max $\bar{x}$ (SD) [Min; Max] | **2.7*** (0.5) | 2.5 (0.5) | 0.136 |
| | | [2.0; 3.0] | [2.0; 3.0] | |
| | TM $\bar{x}$ (SD) [Min; Max] | 0.03 (0.12) | 0.03 (0.14) | 1.000 |
| | | [0.0; 0.7] | [0.0; 0.8] | |
| Untreated caries | DT $\bar{x}$ (IQR) [Min; Max] | **0.7** (1.2) | 0.5 (0.9) | 0.163 |
| | | [0.0; 4.0] | [0.0; 4.0] | |
| Missing teeth | MT $\bar{x}$ (SD) [Min; Max] | 1.0 (1.9) | 1.3 (2.7) | 0.208 |
| | | [0.0; 10.0] | [0.0; 10.0] | |
| Restored teeth | FT Median (IQR) [Min; Max] | **6.0** (4.0) | 4.0 (3.0) | 0.462 |
| | | [0.0; 11.0] | [0.0; 11.0] | |

*p ≤ 0,05 for side comparison between the paretic (PS) and non-paretic side (NPS) of patients with peripheral facial nerve palsy (FPG) using the Wilcoxon-Test regarding the oral health parameters, number of teeth with calculus, approximal plaque index (API), Turesky index (TI), sulcus bleeding index (SBI), papillary bleeding index (PBI), vestibular probing depth (VPD), attachment loss (AL), periodontal screening index (PSI MAX and Median), tooth mobility (TM), number of untreated carious (decayed teeth; DT), number of missing teeth (missed teeth; MT) as well as number of restored teeth (filled teeth; FT), information by means of median and interquartile range (IQR) or mean value ($\bar{x}$) and standard deviation (SD) as well as minimum and maximum [Min; Max] and bilateral asymptotic significance p.

(Table 3). Untreated carious lesions were diagnosed more than twice as often in the FPG compared to the CG (1.3 DT vs. 0.5 DT; p = 0.024; Table 3). Calculus and tooth mobility did not differ between both groups (Table 3).

**Comparison of oral health parameters between paretic and non-paretic side.** The PS revealed significantly higher plaque contamination, more bleeding and greater probing depth, attachment loss and higher PSI values than the NPS (Table 4). Plaque contamination on smooth and approximal surfaces was also higher on the PS than on the NPS (API p = 0.004; TI p ≤ 0.002 Table 4). The median difference in male patients (0.8 vs. 0.5; p = 0.001) exceeded that of female patients (0.6 vs. 0.4; p = 0.002). Within the subgroup analysis, a greater side difference (0.7 vs. 0.5; p ≤ 0.001) was documented in patients with right-sided FP than in patients with left-sided FP (0.5 vs. 0.4; p > 0.05).

Gingival bleeding on probing was significantly higher on the PS than on the NPS (Table 4).

Probing depths were deeper at the PS compared to the NPS and the attachment loss was higher. However, no significance could be demonstrated. The mean PSI reached significantly

higher values on the PS than on the NPS (p = 0.032; Table 4), indicating more affected gingival health. The prevalence of untreated carious lesions was 0.2 DT higher on the PS than on the NPS (Table 4). One individual case impressively illustrates the differences between the caries free NPS compared to the PS with multiple caries lesions.

The PS had a higher number of restorations than the NPS (Table 4). No differences between the PS and the NPS were found concerning tooth mobility and calculus. However, one individual case illustrates more calculus deposits in the PS than in the NPS.

## Self-assessment of oral hygiene and oral hygiene behaviour

Within the FPG, 62.8% (n = 27) of patients maintained their usual cleaning techniques after becoming affected by FP, while 20.9% (n = 9) increased their oral hygiene frequency. Due to subjectively perceived restrictions to oral hygiene, 14.0% (n = 6) of patients avoided the use of dental floss and mouthwashes. Furthermore, patients reported that they switched from electrical (both rotary oscillating electric toothbrushes and sonic toothbrushes)to manual toothbrushes due to changes in sensitivity. More than half of patients (55.8%) believed that they did not have to carry out more intensive oral hygiene on the PS. After FP onset, 18.6% (n = 8) of patients observed more frequent gingival bleeding and 14.0% (n = 6) observed increased tooth sensitivity. Since being affected by paresis, 11.7% (n = 5) noticed increased gingival recession and 9.3% (n = 4) increased tooth mobility. In addition, 55.8% (n = 24) of patients complained of frequent cheek biting, 79.1% (n = 34) frequent food residues in the cheek pouch and 76.7% (n = 33) of fluid loss through the paralysed corner of the mouth.

## Oral health related quality of life

The FPG shows significantly stronger impairment in OHQoL than the CG (Table 5). The median sum of individual responses was 13.0 for the FPG and 1.0 for the CG (Table 5). Group membership had a very strong effect on OHQoL (Cohen's d: 1.4; 95% CI: 0.9–1.9). In both groups, males reflected less restricted OHQoL than females (Table 5). Within the FPG, the

**Table 5. Comparison of OHIP-G14-subscores 1, 3, 5 and total-score between patients with peripheral facial nerve palsy and the control group.**

| OHIP-G14-Subscores | Gender | FPG | CG | p-Value |
|---|---|---|---|---|
| Number | total | 42 | 43 | |
| | male | 14 | 14 | |
| | female | 28 | 29 | |
| **Subscore 1** functional impairement median (IQR) [Min; Max] | total | 1.3 (1.6) [0;4]* | 0.0 (0.0) [0;1] | ≤ **0.001** |
| | male | 0.5 (2.1) [0;3] * | 0.0 (0.0) [0;1] | **0.01** |
| | female | 1.5 (1.5) [0;4] * | 0.0 (0.0) [0;0.5] | ≤ **0.001** |
| **Subscore 3** psychological complaints median (IQR) [Min; Max] | total | 1.4 (2.6) [0;4]* | 0.0 (0.5) [0;2] | ≤ **0.001** |
| | male | 0.5 (2.0) [0;4] | 0.0 (0.1) [0;2] | 0.069 |
| | female | 2.0 (2.9) [0;4] * | 0.0 (0.5) [0;2] | ≤ **0.001** |
| **Subscore 5** mental disability median (IQR) [Min; Max] | total | 1.4 (2.2) [0;4]* | 0.0 (0.0) [0;1.8] | ≤ **0.001** |
| | male | 0.4 (1.4) [0;3,6] | 0.0 (0.2) [0;1.8] | 0.094 |
| | female | 1.8 (1.4) [0;4] * | 0.0 (0.0) [0;1.2] | ≤ **0.001** |
| **OHIP-Total-Score** median (IQR) [Min; Max] | total | 13.0 (17.0) [0;56] * | 1.0 (3.0) [0;2.3] | ≤ **0.001** |
| | male | 5.5 (19.0) [0;37] * | 0.5 (3.0) [0;2.3] | **0.014** |
| | female | 14.0 (16.0) [0;56] * | 1.0 (3.0) [0;10] | ≤ **0.001** |

*p ≤ 0.05 for comparison of OHIP-G14-Subscores 1, 3, 5 and Total-Score between patients with peripheral facial nerve palsy (FPG) and the control group (CG)using Mann-Whitney U-Test, given in number and median with interquartile range as well as minimum and maximum value

difference between males and females was greater than in the CG. However, gender difference in both groups was not significant. In the FPG, high restrictions appeared in subscores 3 (psychological complaints), 5 (psycho-logical handicap) and 1 (functional impairment) (Table 5). Within the subscores, females re-ported higher limitations than males (Table 5).

## Factors influencing oral health

Patient-related influencing factors: duration of disease, disease-related quality of life, etiology of disease and side of facial nerve palsy together with the OHQoL were included in a bivariate correlation analysis according to Spearman and supplemented by effect size calculation (Cohen's d; Table 6). The strongest effect size revealed the disease-related quality of life measured by FDI and FaCE on the OHQoL. The more reduced the disease-related quality of life was rated, the more impairment in OHQoL (Table 6) was reported. In addition, missing teeth on the paretic side correlated with OHQoL significantly, having medium effect size. The duration of paresis did not correlate with impairments in OHQoL (Table 6).

Significant correlating factors were further included in a linear regression analysis. The strong influence of disease-related quality of life on OHQoL remained significant (R = -0.193; p = 0.061).

The effect sizes of oral health parameters on OHQoL were higher in the FPG than in the CG (Table 7). Within the FPG, gender, the number of carious and restored teeth, and the DMFT index correlated significantly with OHQoL (Table 7). Females in the FPG reported a higher limitation of OHQoL than females in the CG.

The patient-related parameters duration of paresis, disease-related quality of life and OHQoL were included in a linear regression analysis together with the oral health parameters of the CG. There were significant regressions between the mean TI and the disease-related quality of life measured by FaCE (b = -0.009; p = 0.004). The lower the FaCE score, the more

**Table 6. Correlation between patient-related factors and oral health-related quality of life.**

| Patient related factors | Correlation (Spearman) [p-value] | ES |
|---|---|---|
| Duration of illness (in months) | -0.132 [0.400] | 0.3 [+] |
| Disease-related quality of life (FaCE) | **-0.653 [< 0.001]** * | 1.7 [+] |
| Disease-related quality of life (FDI) | **-0.690 [< 0.001]** * | 1.9 [+] |
| PS (left/right) | -0.286 [0.068] | 0.6 [+] |
| Etiology of the disease | 0.007 [0.966] | 0.01 |
| Calculus (PS) | -0.140 [0.371] | 0.3 [+] |
| SBI (PS) | -0.216 [0.164] | 0.4 [+] |
| TI (PS) | -0.029 [0.852] | 0.1 |
| TM (PS) | 0.096 [0.450] | 0.2 [+] |
| AL (PS) | 0.050 [0.752] | 0.1 |
| DT (PS) | -0.271 [0.072] | 0.6 [+] |
| MT (PS) | **0.306 [< 0.046]** * | 0.6 [+] |
| FT (PS) | 0.203 [0.192] | 0.4 [+] |
| DMFT (PS) | 0.212 [0.173] | 0.4 [+] |

*p ≤ 0,05 for correlation analysis according to Spearman for the parameters Duration of illness, Disease-related quality of life (Facial Clinimetric Evaluation and Facial Disability Index), paretic side (PS), Etiology of the disease, calculus, sulcus bleeding index (SBI), Turesky index (TI), tooth mobility (TM), attachment loss (AL), decayed teeth (DT), missed teeth (MT), filled teeth (FT), decayed-missed-filled teeth index (dmft-index)

[+] cohen' s d ≥0,3

**Table 7. Correlation between patient-related factors and OHIP-total-score of the patients with peripheral facial nerve palsy and the control group.**

| Patient related factors | FPG | | CG | |
| --- | --- | --- | --- | --- |
| | Correlation (Spearman) [p-value] | ES | Correlation (Spearman) [p-value] | ES |
| age | 0.137 [0.381] | 0.3 + | -0.039 [0.804] | 0,1 |
| gender | **0.304 [0.047]** * | 0.6 + | -0.002 [0.989] | 0,0 |
| Calculus (entire dentition) | -0.141 [0.366] | 0.3 + | 0.050 [0.748] | 0,1 |
| SBI (entire dentition) | -0.255 [0.099] | 0.5 + | 0.111 [0.479] | **0,2** |
| TI (entire dentition) | -0.004 [0.977] | 0.0 | 0.143 [0.326] | **0,3** |
| TM (entire dentition) | 0.038 [0.808] | 0.1 | 0.093 [0.554] | 0,2 |
| AL (entire dentition) | 0.027 [0.865] | 0.1 | -0.030 [0.847] | 0,1 |
| DT (entire dentition) | **-0.336 [0.027]** * | 0.7 + | 0.078 [0.620] | 0,2 |
| MT (entire dentition) | 0.244 [0.114] | 0.5 + | -0.084 [0.592] | 0,2 |
| FT (entire dentition) | **0.349 [0.022]** * | 0.7 + | -0.093 [0.554] | 0,2 |
| DMFT (entire dentition) | **0.296 [0.054]** * | 0.6 + | -0.066 [0.676] | 0,1 |

*p ≤ 0,05 for correlation analysis according to Spearman for the parameters age, gender, calculus, sulcus bleeding index (SBI), Turesky index (TI), tooth mobility (TM), attachment loss (AL), decayed teeth (DT), missed teeth (MT), filled teeth (FT), decayed-missed-filled teeth index (dmft-index) between patients with peripheral facial nerve palsy (FPG) and the control group (CG)

+effect size Cohen's d ≥0.3

plaque was identified covering the tooth surfaces. Furthermore, a significant regression of the MT and the disease-related quality of life measured by FDI score (b = -0.098; p = 0.003) was determined. In patients with lower FDI scores, more missing teeth were diagnosed. There was a significant positive correlation between the FDI and the duration of illness in months (R = 0.374; p = 0.013). The perceived limitations in the disease-related quality of life decreased with longer lasting illness.

Other influencing factors on oral health, such as the frequency of use of certain oral care aids or the cleaning technique used, were examined using a bivariate correlation analysis according to Spearman (Table 8). The use of electric toothbrushes resulted in significantly less calculus and plaque. More frequent use of dental floss was associated with lower approximal and general plaque contamination and better periodontal health. The use of interdental brushes had no significant impact on plaque and oral health. Unsystematic tooth brushing fostered gingival bleeding. A higher prevalence of untreated carious lesions and reduced periodontal health were associated with smoking. Patients who reported no loss of fluid from the drooping corner of the mouth of the PS had more calculus than patients who reported frequent fluid leakage. Patients who noted food debris remaining in the cheek pouches revealed significantly lower proximal plaque, but significantly higher DMFT and more restored teeth. Patients feeling less attractive had more restored or missing teeth. Patients rating their own oral hygiene as poor had more dental plaque (Table 8).

In a multivariate linear regression analysis, use of an electric toothbrush (p < 0.003) and the loss of fluid (p = 0.001) on calculus, the influence of unsystematic tooth brushing on the SBI (p = 0.010), as well as the interaction between carious lesions and smoking behaviour (p = 0.007), were found to be significant.

## Discussion

In this case-control-study, oral health, oral hygiene behavior and oral health related quality of life in patients with chronic peripheral facial nerve palsy compared to healthy controls was examined.

Table 8. Bivariate correlation analysis according to Spearman, factors influencing oral health on the paretic side of patients with peripheral facial nerve palsy (R [p-value]).

| Influencing factors | Importance of tooth preservation (+) | Electric toothbrush (++) | Manual toothbrush (++) | Floss (++) | Interdental space brushes (++) | Frequent use (+++) | Unsystematic cleaning (+++) | Smoking behaviour (+++) | Fluid leakage (+++) | Food debris in cheek (+++) | Attractiveness (+) | Assessment of oral hygiene (+) | Number of visits to the dentist (+) |
|---|---|---|---|---|---|---|---|---|---|---|---|---|---|
| Calculus | 0.125 [0.425] | **-0.428*** **[0.004]** | 0.229 [0.139] | -0.174 [0.264] | 0.054 [0.729] | 0.015 [0.923] | 0.006 [0.970] | -0.044 [0.779] | **0.422*** **[0.005]** | 0.047 [0.765] | -0.006 [0.970] | 0.229 [0.140] | -0.030 [0.846] |
| API (%) | 0.085 [0.588] | **-0.338*** **[0.027]** | 0.116 [0.460] | **-0.338*** **[0.027]** | -0.014 [0.930] | 0.082 [0.685] | -0.190 [0.229] | -0.140 [0.369] | 0.062 [0.691] | **0.322*** **[0.035]** | 0.182 [0.243] | **0.371*** **[0.014]** | 0.137 [0.382] |
| TI | 0.085 [0.588] | **-0.377*** **[0.013]** | 0.235 [0.129] | **-0.361*** **[0.017]** | -0.017 [0.914] | 0.275 [0.078] | 0.287 [0.065] | -0.224 [0.149] | 0.016 [0.921] | 0.113 [0.471] | 0.002 [0.988] | 0.065 [0.631] | -0.055 [0.725] |
| SBI (%) | -0.095 [0.554] | -0.052 [0.741] | -0.087 [0.581] | -0.154 [0.324] | 0.083 [0.596] | -0.091 [0.568] | **0.478*** **[<0.001]** | -0.094 [0.548] | 0.205 [0.188] | 0.196 [0.207] | 0.006 [0.971] | 0.096 [0.541] | -0.105 [0.503] |
| PBI (%) | -0.089 [0.570] | 0.015 [0.926] | 0.026 [0.867] | 0.160 [0.305] | 0.085 [0.587] | -0.075 [0.638] | 0.138 [0.384] | -0.219 [0.158] | 0.149 [0.341] | 0.159 [0.308] | 0.131 [0.403] | 0.174 [0.263] | -0.234 [0.132] |
| PSI | -0.071 [0.650] | 0.237 [0.126] | **0.349*** **[0.022]** | **-0.341*** **[0.025]** | 0.295 [0.055] | 0.072 [0.652] | 0.037 [0.814] | **-0.301*** **[0.050]** | -0.136 [0.383] | -0.083 [0.597] | 0.126 [0.422] | 0.117 [0.453] | -0.066 [0.674] |
| DT | -0.241 [0.120] | 0.001 [0.994] | 0.048 [0.759] | -0.236 [0.127] | -0.042 [0.790] | -0.020 [0.900] | -0.051 [0.750] | **-0.588*** **[<0.001]** | 0.118 [0.452] | 0.095 [0.544] | -0.296 [0.054] | 0.270 [0.080] | 0.230 [0.138] |
| MT | 0.159 [0.309] | -0.165 [0.290] | 0.248 [0.110] | -0.172 [0.271] | 0.048 [0.758] | 0.156 [0.322] | 0.009 [0.955] | 0.134 [0.391] | -0.073 [0.641] | -0.265 [0.097] | **0.344*** **[0.024]** | -0.127 [0.415] | -0.249 [0.108] |
| FT | 0.030 [0.847] | -0.072 [0.647] | -0.005 [0.973] | 0.115 [0.462] | 0.225 [0.148] | -0.113 [0.476] | -0.113 [0.475] | 0.190 [0.233] | 0.233 [0.133] | **-0.289** **[0.061]** | 0.287 [0.065] | -0.001 [0.007] | 0.023 [0.885] |
| DMFT | -0.010 [0.951] | 0.152 [0.329] | 0.140 [0.369] | -0.090 [0.567] | 0.116 [0.458] | -0.005 [0.975] | -0.046 [0.775] | 0.061 [0.700] | 0.150 [0.338] | **-0.331*** **[0.030]** | **0.338*** **[0.027]** | 0.031 [0.845] | -0.040 [0.799] |

* $p \leq 0.05$; for bivariate correlation analysis according to Spearman for factors influencing oral health on the paretic side of patients with peripheral facial nerve palsy regarding the oral health parameters number of teeth with calculus, approximal plaque index (API), Turesky index (TI), sulcus bleeding index (SBI), papillary bleeding index (PBI), periodontal screening index (PSI), number of untreated carious (decayed teeth; DT), number of missing teeth (missed teeth; MT) as well as number of restored teeth(filled teeth; FT), decayed-missed-filled teeth index (dmft), information by means of median and interquartile range (IQR) or mean value ($\bar{x}$) and standard deviation (SD) as well as minimum and maximum [Min; Max] and bilateral asymptotic significance p; + metrical; ++ ordinal; +++ nominal (different scaling of the influencing factors)

## Oral health

The results of this study revealed significant differences in oral health between participants with and without FP and between the paretic and non-paretic side (PS; NPS). The plaque contamination of the FPG exceeded that of the CG, which was demonstrated by a significantly higher API (53.6% vs. 39.9%) and TI (0.6 vs. 0.4). Consistent with these results, a case study [21] described poor oral hygiene with excessive plaque and calculus contamination in a girl with left-sided FP. A broad overview describing oral health and hygiene status of patients with FP–to the best of our knowledge–is not available in the literature, limiting the comparison of our findings. The present study contributes to closing this evidence gap and to raising the awareness of oral health problems in patients with FP. The dentition at the patient's PS was significantly more infested with dental plaque than at the NPS. This observation contrasts a case report describing an equal distribution of dental plaque on both, the PS and the NPS [22]. A significantly higher bleeding tendency was diagnosed in the FPG than in the CG. The prevalence of sulcus bleeding in the CG is consistent with representative data from the 5th German oral health study [23]. Sulcus bleeding, indicating gingival inflammation, was recorded significantly more frequently on a patient's PS than on the NPS. This coincides with observations from a case report [22] of a patient with right-sided FP. The present study con-firms, that higher bleeding tendency in the dentition´s PS compared to the NPS is not an singular but an apparently common finding.

As the assessed periodontal health parameters indicate, FP contributes to periodontal health deterioration. Corresponding to our findings, a case report [22] of a patient with right-sided FP de-scribes severe chronic periodontitis, enormous probing depths in the molar region and considerable bone loss at the PS versus the NPS. Zavarella et al. [22] justified this by the lack of nerve supply to the paretic side of the mouth and concluded that the periodontal innervation can contribute to the regulation of local processes that are involved in the pathogenesis of periodontitis.

The incidence of caries was twice as high in the FPG as in the CG (1.3 vs. 0.5). The values of the CG are in line with representative data from the 5th German oral health study [23] (younger adults / younger seniors DT: 0.5). Dental caries in the FPG was even higher than in patients with cerebral palsy, a more severe disease than FP. In a study [24] consisting of 487 patients with cerebral palsy, the prevalence of caries in the over 56 age group was DT = 0.82, followed by the 36 to 55year-olds (DT = 0.76).

The FPG had more missing teeth than the CG, which can be explained by the increased plaque load and consequent caries development. In addition, comparing PS and NPS, more restored teeth were detected on the PS. This underlines the increased dental caries treatment need.

It is conceivable that the poorer health of patients with FP is the result of the lack of targeted muscle movements involved in natural cleaning mechanisms of the oral cavity. A study [6] on 14 patients with acute idiopathic FP reported that the ability to clean the vestibule was significantly lower compared to healthy controls and on their PS compared to their NPS (p ≤ 0.001). Another study confirmed these findings [25]. Pani et al. [26], in a study of 45 adolescents with cerebral palsy, observed that very severe motor impairments lead to significantly worse gingival indices and DMFT values than lower motor limitations. Defective muscle activity results in an accumulation of plaque, which is particularly concentrated on the PS. Increased plaque contamination correlates with increased bleeding tendency. The persistent irritation of the gingiva and interdental papillae as a result of insufficient plaque removal induces chronic inflammation and bleeding more quickly [27]. Significantly increased oral plaque contamination also promotes the development of carious lesions [28, 29]. This could be compensated by

more intensive and effective cleaning of the oral cavity. In the present studyhowever, 55.8% of the patients reported not to have intensified oral hygiene after becoming affected by FP. As a consequence, gingival inflammation and dental caries developed. Additionally, damage to the facial nerve in the facial nerve canal before the exit of the notochord might lead to a dry mouth [30]. Saliva ensures further natural self-cleaning of the oral cavity [30]. A lower saliva flow rate is associated with increased plaque formation and consequently an increased risk of gingival inflammation and caries [30].

Within this study, the periodontal health of patients with facial paresis was not yet significantly worse than that of healthy control subjects. However, the tendency towards decreased oral health existed and the initial symptoms, such as increased plaque infestation and an increased bleeding tendency, could already significantly be detected. The symptoms of FP contribute to vulnerability towards oral conditions and to more difficult circumstances for conducting effective oral hygiene. Consequently, patients with FP are at higher risk of developing dental caries and periodontal diseases. In summary, our first and second null hypotheses can be rejected as explained and instead the corresponding alternative hypotheses can be proven.

## Oral health related quality of life

The first null hypothesis can also be rejected with regard to oral health-related quality of life of the patients with FP compared to our controls. Within the present study, the FPG indicated a lower OHQoL than the CG. Stroke patients also reported a significantly lower OHQoL (OHIP-Total-Score 18.8 ± 15.5) than healthy controls [31]. Based on the results of this study, patients with FP feel similarly severely restricted in their OHQoL as patients after a stroke. However, while the subdomains "functional impairment" and "physical pain" predominated in stroke patients, in the patients with FP the areas "psychological complaints" and "mental disability" were dominant. Dawidjan [24] also described the psychological limitations of patients with FP, which might reduce motivation and lead to neglect of oral health.

The OHQoL of the CG is in line with 60% of the general German population [32]. In the FPG, impairment of OHQoL is higher than in 80% of the general German population. Patients in the FPG frequently complained about "withdrawal from society, especially when eating", describing a loss of social contact. Furthermore, they often suffer from constant pain due to temporomandibular joint tension caused by overloading the NPS. In addition, due to the altered activity of the lip muscles, sexual restrictions have been described affecting overall OHQoL.

## Oral hygiene behaviour

Restricted muscle activity and altered sensitivity limits a patient's use of certain oral care aids. This study revealed that patients with FP were less likely to useelectric toothbrushes several times a day, and dental floss and interdental brushes daily than those without FP. If the handling of dental floss is generally considered to be difficult—even without FP—[33], patients with FP are additionally challenged by the limited flexibility of their cheek, according to the patients' testimony. In addition, incomplete lip closure impedes holding fluids in the mouth [3]. This explains problems associated with rinsing after tooth cleaning, spitting out toothpaste or using mouthwashes. However, the present study revealed that the use of an electric toothbrush and the more frequent use of dental floss lead to significantly less calculus and plaque contamination and better periodontal health. Therefore, these oral care aids should be particularly recommended to patients and individual training offered, despite the difficulties mentioned. Bozkurt et al. [34] recommends electric toothbrushes to neuromuscular disabled patients. The superiority of electric toothbrushes compared to manual brushes in patients with

gingivitis and periodontitis was described in a systematic review [35]. The development of oral care utensils specially tailored to patients with FP, such as special brush heads for manual or electric toothbrushes, which on the one side cleans the teeth and on the other side massages the cheek muscles to promote flexibility, would be conceivable. The successful use of specially designed or adapted toothbrushes in patients with cerebral palsy has beenreported [36]. In the literature [37], temporary tooth spacers on the lateral side of the molars were also recommended to protect against bite wounds during mastication. In addition, toothpastes with a reduced proportion of foaming agent and a reduced amount of toothpaste could be recommended in order to minimise the unpleasant loss of fluid through the corners of the mouth of the PS.

In the present study, unsystematic oral hygiene behaviour was documented more frequently in the FPG than in the CG. The majority of patients paid no special attention to the PS in relation to tooth cleaning, which was also described in a case report [22]. One patient stated that "there are enough other problems caused by the paresis" and that he "does not want to put an additional emphasis on oral hygiene on the PS". It would be advisable to explain and offer a systematic, individually adapted cleaning routine to the patient. In addition, patients in the FPG visited their dentists less often than patients in the CG (twice per year: 53.5% vs. 60.5%), were more likely to visit only when they had complaints 4.7% vs. 2.3%, and were more likely to not have visited their dentist for many years: 2.3% vs. 0.0%. One can assume that the reason for this is, on the one hand, the psychological stress experienced by the patient and, on the other, the high expenditure of time due to other frequent medical appointments and facial muscle training. Nevertheless, in addition to the various problems that patients with FP are experiencing, special attention should be paid to oral hygiene. Interestingly, patients who noted food debris remaining in their cheek pouches tended to intensify tooth brushing. Their plaque and bleeding was consequently lower than in patients who did not notice this problem. The latter were possibly less sensitised to their oral situation, which should be taken into account in the future treatment of patients with FP. All in all, our third null hypothesis can also be rejected in conclusion.

Good oral hygiene contributes significantly to wellbeing, self-esteem and social acceptance [38]. Therefore, it is important to provide dental care to patients with FP and to prevent their oral hygiene from deteriorating as a result of FP. In order to start prevention at an early stage, we recommend interdisciplinary collaboration in the treatment of patients with FP, as do Aranka et al. [39]. Oral health and hygiene should receive the same attention as other important consequences of FP, such as reduced aesthetics [40] or eye complaints [41].

### Factors influencing oral health and oral hygiene behaviour

With regard to our fourth null hypothesis, we can summarise the following. Within the subgroup analysis in the present study, a greater difference in plaque contamination between both sides of the mouth was documented in patients with right-sided FP than in the patients with left-sided FP (right-sided FP: 0.7 vs. 0.5; $p \leq 0.001$; left-sided FP: 0.5 vs. 0.4; $p > 0.05$; $p = 0.004$). A patient's handedness might explain these differences. While in patients with left-sided FP the paresis was on the side of the mouth that is easier to clean for right-handers, right-handed patients with right-sided FP were exposed to double stress [42]. However, within a multiple linear regression analysis, this correlation could not be proven to be significant ($p = 0.541$). Nevertheless, we recommend taking this aspect into account when designing individualised toothbrushing training. However, the patients with left-sided disease were on average 13.5 years younger than the patients with right-sided disease (mean age of patients with left-sided disease: 41.5 years vs. mean age of patients with right-sided disease: 55.0 years), so

that the difference between the paretic and non-paretic halves of the mouth could also be due to poorer handling of the patients with right-sided disease during tooth cleaning. However, the age difference between right- and left-sided patients also has no significant influence on the difference between the two groups (p = 0.540).

With regard to the disease-related quality of life, it was found that the lower the FaCE score, the more plaque covered the tooth surfaces. Additionally, in patients with lower FDI scores more missing teeth were diagnosed. These relationships were significant with a p-value of p = 0.01. Thus, it can be said that the patient-related parameter of disease-related quality of life influences oral health, although not with respect to all parameters. With regard to the duration of the paresis, no significant relationships with oral health and oral hygiene behaviour were found. Thus, it is not possible to clearly determine within patients suffering from facial nerve palsy whether the risk of oral health impairments is more likely to affect patients with the disease for a shorter or longer period of time. Therefore, it is important to take prophylactic dental measures in all patients with facial nerve palsy. In summary, the fourth null hypothesis can only be partially rejected. The patient-related parameters of handedness and disease-related quality of life have a partial influence on the oral health of the patients, but no significant relationships could be demonstrated with regard to the duration of paresis.

### Prevention suggestions and advice for patients

To create a holistic therapy concept for patients with FP, we recommend including dentists in the interdisciplinary team. Medical staff of all disciplines involved in the treatment of FP and dental staff shall cooperate to reduce the double burden of disease. After diagnosis, patients should initially visit the dentist every 3 months to benefit from intensive preventive care, professional dental cleaning, targeted oral hygiene instructions and individualised oral hygiene training. Furthermore, it is important to alert patients to oral deficits and motivate them to focus on dental and oral care. Special attention should be paid to target muscle training and special muscle massage before and after tooth brushing because most patients with synkineses have a very stiff cheek. Thereby patients might relax muscles and assure enough space in their mouth for dental care aids. For example, patients can stretch their cheek, loosen and blow up cheeks or use the vibration of the electric toothbrush to massage the muscles. In order to be able to perform the exercises correctly, it is recommend to consult a specialist facial therapist. Patients should be advised not to spare the paretic side, to adopt a systematic cleaning routine and brush their teeth after every meal and before going to sleep. With regards to the above-mentioned advice, a flyer has been prepared that can be given to patients at the time of diagnosis (S2 File. Flyer: good oral health despite facial palsy).

### Strengths and limitations of this study

To avoid confounding and handling bias, we matched patients and controls by age and gender which makes both groups comparable. Furthermore, we conducted a standardised, objectified examination, following the same protocol for each patient and calibrated the examining dentist before the start of the study through an examination training (see 2.2 Calibration of the examiner). Based on evaluation using kappa-statistic, the examiner (L. S.) achieved an agreement of 0.86 with respect to the DMFT. According to Landis/Koch, this corresponds to a strong agreement. With regard to the Turesky Index, a substantial agreement of 0.75 was found between the two examiners In addition, we examined only consecutive patients who appeared randomly at the center. Thus, a random selection was made to avoid a selection bias. Furthermore, to exclude confounding bias as far as possible, we included, beside the inspection of oral health, a self-management and the OHQoL and excluded specific diseases To reduce

the problem of multiple testing, we corrected our values with the Bonferroni correction. Therefore, we distinguished seven independent hypotheses. The individual parameters within these hypotheses are interconnected and investigated the same research question so that not each parameter has to be corrected individually, but only the value of the respective hypothesis. Including only patients with FP increased the homogeneity in the FPG so that oral health was observed in a defined symptom group. All patients were examined under the same conditions and according to the same procedure by the same calibrated examiner. Calibration of the single examiner reduced observation bias. Conducting the study in the specialised facial nerve center enabled the assessment of patients with FP from across Germany which is a significant strength of the study. Thus, the study provides good generalisability and transferability and is not limited to the regional setting.

Nevertheless, the study has some limitations. Patients were examined in the hospital ward and not in a dental clinic. This made the collection of oral health parameters more difficult. More convenient for future studies would be the ability to transfer patients from the hospital to a dental office for oral examination. A major limitation of the study design is the lack of blinding. Due to the nature of the FP, it was not possible to blind the examiner with regard to the group affiliation of the patients.

## Conclusion

Patients with FP have poorer oral health and oral hygiene, especially on the side of the face affected by FP, and poorer OHQoL than patients without FP. Individual oral hygiene behaviour as well as patient-related parameters such as handedness or disease-related quality of life influence the oral health of patients. Adding oral health care to the existing therapy guidelines [7] and establishing an interdisciplinary cooperation might prevent oral health from being neglected producing unpleasant consequences. We therefore recommend that patients be referred to a dentist in addition to the other disciplines once a diagnosis has been made.

## Supporting information

**S1 Checklist. STROBE checklist.**
(DOCX)

**S2 Checklist. TREND statement checklist.**
(DOCX)

**S1 File. Minimum data set of patient (FPG) and controls (CG).** All raw data collected during the study implementation between the 01.09.2020 and 14.10.2020 are available in this excel file.
(XLSX)

**S2 File. Flyer: Good oral health despite facial nerve palsy.** We developed a flyer with helpful tips and advice for patients with chronic facial palsy. The file is in Word format and can be modified accordingly. The flyer can be given to the patients after diagnosis in order to maintain the oral health of the affected patients in the long term.
(DOCX)

**S3 File.**
(DOCX)

**S4 File.**
(DOCX)

## Acknowledgments

We thank C. Kaden for his technical support and for helping us with Excel. Additionally, we thank X. Müller, M. Bober-Reeves and T. Bober for correcting the English language and expression.

## Author Contributions

**Conceptualization:** Lisa Strobelt.

**Data curation:** Lisa Strobelt.

**Formal analysis:** Lisa Strobelt, Thomas Lehmann.

**Investigation:** Lisa Strobelt.

**Methodology:** Lisa Strobelt.

**Project administration:** Gerd Fabian Volk, Ina Manuela Schüler.

**Software:** Thomas Lehmann.

**Supervision:** Anna-Maria Kuttenreich, Gerd Fabian Volk, Ina Manuela Schüler.

**Validation:** Anna-Maria Kuttenreich, Gerd Fabian Volk, Ina Manuela Schüler.

**Visualization:** Lisa Strobelt.

**Writing – original draft:** Lisa Strobelt.

**Writing – review & editing:** Carien Beurskens.

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
