## [Decision Letter · Decision Letter 0]

26 Apr 2022

PONE-D-21-21348Oral health and oral health-related quality of life in patients with chronic peripheral facial nerve palsy with synkinesis- a case-control-studyPLOS ONE

Dear Dr. Strobelt,

Thank you for submitting your manuscript to PLOS ONE. I sincerely apologise for the unusually delayed timeframe for reviewing your manuscript. After careful consideration, we feel that it has merit but does not fully meet PLOS ONE’s publication criteria as it currently stands. Therefore, we invite you to submit a revised version of the manuscript that addresses the points raised during the review process. Your manuscript has been evaluated by a reviewer with expertise in biostatistics and clinical research study design. Although they find the study to be very interesting that it addresses an important topic, they nonetheless have several recommendations for further improving the methodology and reporting. Please ensure that their comments are addressed in full. Please note that we have only been able to secure a single reviewer to assess your manuscript. We are issuing a decision on your manuscript at this point to prevent further delays in the evaluation of your manuscript. Please be aware that the editor who handles your revised manuscript might find it necessary to invite additional reviewers to assess this work once the revised manuscript is submitted. However, we will aim to proceed on the basis of this single review if possible.

We look forward to receiving your revised manuscript.

Kind regards,

Emily Chenette

Editor in Chief

PLOS ONE

Journal Requirements:

3. Please ensure that you refer to Figure 6 in your text as, if accepted, production will need this reference to link the reader to the figure.

4. We note you have included a table to which you do not refer in the text of your manuscript. Please ensure that you refer to Table 4 in your text; if accepted, production will need this reference to link the reader to the Table.

5. Please upload a copy of Supporting Information S1 Table 1, S2 Table 2, S3 Table 3, S4 Fig 1, S5 Fig 2, S6 Fig 3A, S7 Fig 4a and b, S8 Fig 5 a and b, S9 Table 4, S10 Table 5, S11 Table 6, S12, Table 7, S13 Table 8,  which you refer to in your text on pages 25 and 26.

Reviewers' comments:

Reviewer's Responses to Questions

**Comments to the Author**

1. Is the manuscript technically sound, and do the data support the conclusions?

Reviewer #1: No

2. Has the statistical analysis been performed appropriately and rigorously? 

Reviewer #1: No

3. Have the authors made all data underlying the findings in their manuscript fully available?

Reviewer #1: No

4. Is the manuscript presented in an intelligible fashion and written in standard English?

Reviewer #1: Yes

5. Review Comments to the Author

Reviewer #1: This is a very interesting randomised case-control study examining oral health in patients with facial never palsy with synkinesis. Well done to team for carrying out such important research.

The major comment is, this is a case-control study, and the authors could really benefit from using the STROBE guidelines for reporting. The manuscript could benefit with some restructuring of presentation of all sections to increase readability.

They are some comments worth mentioning for the authors attention.

1) Very confusing the title says case-control study but a CONSORT has been presented relating to an RCT study design and you have a level labelled ‘randomised’ in the CONSORT. You can have a flow diagram showing the cases and controls and flow of participation throughout the study.

2) Remove the percentages from CONSORT as its confusing, absolute numbers will suffice and just label the figure a flow diagram

3) Authors should include details of how individuals were matched, i.e number of cases per control. Also for an example were people matched on exact age or age range?

4) Authors should give more information on the sample size as it is difficult to replicate, i.e what effect was planned to be detected with the anticipated sample size. e,g anticipated effect in control group etc..

5) Was data anonymised, i.e. patients assigned an identifier number.

6) Data recorded with Microsoft Excel, more details about data security should be given, also who entered the data.

7) “To check the significance, the Mann-Whitney-U-test was used in the case of two independent samples and the Wilcoxon test in the case of two paired samples.” – This sentence doesn’t make sense.

8) Have a section or include a section to make it distinct what the primary outcome. i.e oral health, its definition (looks like it’s a composition of several parameters) and details of how/when its measured, assuming collected at study enrolment.

9) Statistical analysis section needs to be expanded further, as pertinent details such as confounding, handling bias, handling missing data etc, checking model assumption are missing should be stated.

10) Table 2 should probably be stratified by cases and controls and and an overall column.

11) Medians should be reported with IQRs.

12) With so many tests was the p-value adjusted for multiple testing.

13) Table 1 can probably be written as a text.

14) "A total of 86 people aged between 21 and 77 years, 58 females (67.4%), were examined (Table 1). The median age was 51.0 years." Should this Table 2? And the sentence re median age, which group and also cold benefit from giving IQRs.

15) Authors should label male, female in the tables instead of signs.

6. PLOS authors have the option to publish the peer review history of their article (what does this mean?). If published, this will include your full peer review and any attached files.

Reviewer #1: No

---

## [Author Response · Author response to Decision Letter 0]

3 Jun 2022

Emily Chenette

Editor in Chief

PLOS ONE

Rebuttal to the reviewer

Manuscript: “Oral health and oral health-related quality of life in patients with chronic peripheral facial nerve palsy with synkineses - a case-control-study”

Ref.: PONE-D-21-21348

Dear Editor,

We greatly appreciate the comments and the reviews of our manuscript submitted for publication in the prestigious journal Plos One. We are pleased to note that you feel that the paper is interesting and relevant in its field. The comments were helpful in identifying areas for improvement, clarification or addition of further details. 

In this letter we provide point-by-point responses the reviewers’ comments and explain all text and table changes made to the manuscript.

Due to the prolonged review process, we have used the time to work out therapeutic recommendations to avoid or reduce the oral health deficits, we have found in this publication. For this, we have cooperated closely we Carien Beurskens, who has many years of experience with patients with facial palsy. Her very valuable contribution to this paper can be mainly found in chapter 4.4 (Preventions suggestions and advices for patients) and in the Supporting Information. In addition, she also helped us to improve the whole manuscript and our English language. These contributions justify and deserves a co-authorship of her. During the revision process, Thomas Lehmann also made a significant contribution to improving statistical matters. Since he has provided statistical support to the study from the beginning, he also deserves to be included as a co-author.

We look forward to your favorable review and remain at your disposal should any further information be required.

With many thanks on behalf of all co-authors,

Yours sincerely,

Lisa Strobelt 

General remarks of the authors

We are grateful for the comments of the reviewer. All comments were carefully considered by all co-authors and incorporated into this new draft of the manuscript. The feedback allowed us to clarify important aspects of the study. We are convinced that the revised manuscript presented herewith is improved over the previous version. This letter will address all comments made in full detail and explain the respective changes we have made to the text.

1. Reply to Journal Requirements

1.1 Concern of the Journal:

Our response:

We have read through your formatting guidelines again and implemented the style and file naming requirements in the revised manuscript. In the process we changed the figure designations from “figures” to “figs”, the single-space paragraph format in double-space paragraph format, exclude physical addresses and list corresponding author’s initials in parentheses after the email address on the title page.

1.2 Concern of the Journal:

In your Data Availability statement, you have not specified where the minimal data set underlying the results described in your manuscript can be found. PLOS defines a study's minimal data set as the underlying data used to reach the conclusions drawn in the manuscript and any additional data required to replicate the reported study findings in their entirety. All PLOS journals require that the minimal data set be made fully available. For more information about our data policy, please see http://journals.plos.org/plosone/s/data-availability.

Our response:

Thank you very much for this advice. We have now made our minimal data set available under S1 File. Minimum data set of patients (FPG) and controls (CG). It can be found in an uploaded excel table. 

Revised text:

S1_File. Minimum data set of patient (FPG) and controls (CG). All raw data collected during the study implementation between the 01.09.2020 and 14.10.2020 are available in this excel file.

1.3 Concern of the Journal:

Please ensure that you refer to Figure 6 in your text as, if accepted, production will need this reference to link the reader to the figure.

Our response:

Thank you very much for drawing our attention to the incorrect numbering. We have changed the image numbering in the text and refer to Figure 6.

1.4 Concern of the Journal:

We note you have included a table to which you do not refer in the text of your manuscript. Please ensure that you refer to Table 4 in your text; if accepted, production will need this reference to link the reader to the Table.

Our response:

Thank you very much for drawing our attention to the incorrect numbering. We have changed the table numbering in the text and refer Table 4.

1.5 Concern of the Journal:

Please upload a copy of Supporting Information S1 Table 1, S2 Table 2, S3 Table 3, S4 Fig 1, S5 Fig 2, S6 Fig 3A, S7 Fig 4a and b, S8 Fig 5 a and b, S9 Table 4, S10 Table 5, S11 Table 6, S12, Table 7, S13 Table 8, which you refer to in your text on pages 25 and 26.

Our response:

We made a mistake on this point during the submission of the first manuscript. This information is only an overview of the images and tables included in the manuscript. We apologize for this confusion. 

We have now correctly listed the supporting information we would like to provide.

2. Reply to reviewer #1

2.1 Concern of the Reviewer:

This is a very interesting randomised case-control study examining oral health in patients with facial nerve palsy with synkineses. Well done to team for carrying out such important research.

Our response:

Thank you very much for this appreciating comment. We are very pleased that you praise our research work to such an extent.

2.2 Concern of the reviewer: 

The major comment is, this is a case-control study, and the authors could really benefit from using the STROBE guidelines for reporting. The manuscript could benefit with some restructuring of presentation of all sections to increase readability.

Our response:

Thank you very much for this advice. We used the STROBE guidelines for our study and integrated it into the manuscript as supporting information (S2 Table. STROBE checklist). In the text we made the following changes in relation to the STROBE guidelines:

Regarding the Item no. 9 Bias, we added a paragraph on page 6/7 and 21/22. 

Bias 9 Describe any efforts to address potential sources of bias 6-7; 21-22

We also integrated the flowchart Fig. 1 into the text and inserted it in the 2.1 Recruitment and Inclusion and Exclusion Criteria section to better follow the order of the STROBE guidelines.

We also defined the main outcome and separated it from other outcomes.

Main results 16 b) Report category boundaries when continuous variables were categorized 7-11

 (c) If relevant, consider translating estimates of relative risk into absolute risk for a meaningful time period not applicable

Other analyses 17 Report other analyses done—eg analyses of subgroups and interactions, and sensitivity analyses 11-16

2.3 Concern of the reviewer:

They are some comments worth mentioning for the authors attention.

Very confusing the title says case-control study but a CONSORT has been presented relating to an RCT study design and you have a level labelled ‘randomised’ in the CONSORT. You can have a flow diagram showing the cases and controls and flow of participation throughout the study.

Our response:

Thank you for this observation. We updated the flowchart to illustrate cases/controls and the study flow. 

2.4 Concern of the reviewer:

Remove the percentages from CONSORT as its confusing, absolute numbers will suffice and just label the figure a flow diagram

Our response:

Many thanks for this suggestion. We revised the content of the flow chart and removed the percentages to avoid confusion. We also removed the designation “CONSORT”.

Revised text:

Fig. 1: flowchart study flow

2.5 Concern of the reviewer:

Authors should include details of how individuals were matched, i.e number of cases per control. Also for an example were people matched on exact age or age range?

Our response:

Thank you very much for this comment. We added that an age range of plus minus 5 years was applied to the match parameter age in order to find suitable controls for our patients. 

Revised text:

Patients with FP (n=43) in the FP-group (FPG) were matched by age and gender with 43 participants without ever diagnosed FP in the control group (CG). With regard to the match parameter age, an age range of 5 years was applied.

2.6 Concern of the reviewer:

Authors should give more information on the sample size as it is difficult to replicate, i.e what effect was planned to be detected with the anticipated sample size. e.g anticipated effect in control group etc..

Our response:

Thank you for this suggestion for improvement. We integrated the values of the sample size calculation from G*power into the manuscript.

Revised text:

To detect an effect size of Cohen’s d=0.67 with a power of 80% in total 86 patients (43 per group) need to be analysed (G*Power 3.1, two sided Wilcoxon-Mann-Whitney Test, significance level 5%; Figure 1).

2.7 Concern of the reviewer:

Was data anonymised, i.e. patients assigned an identifier number.

Our response:

Yes, a pseudonymised recording of the patient/subject data takes place. An individual pseudonym was used for each patient/subject, through which alone the identity of the subject cannot be recognised. A patient/proband identification list was kept. In this list, the patient/proband identification number was linked to the participant's full name, date of birth, age, occupation and, in the case of patients, to the form, severity, duration and cause of the facial nerve palsy in order to enable subsequent identification of the subjects. This document was kept absolutely confidential and is not accessible to third parties.

Revised Text:

A pseudonymised data collection was carried out with the help of a patients/controls identification list. Each patient was assigned an identification number. This list was stored in accordance with data protection requirements and all data analyses were only carried out with the pseudonymised data records.

2.8 Concern of the reviewer:

Data recorded with Microsoft Excel, more details about data security should be given, also who entered the data.

Our response:

Thank you very much for this advice. We added information on data security. The data was stored on a password protected data device to which third parties had no access. No one but the authors could access the data. 

The data were collected and documented by the dentist who conducted the oral health examination.

Revised Text:

The paper-based examination sheets and questionnaires were digitalized, data recorded with Microsoft Excel® (Version 2007, Microsoft Corp.) and analyzed with IBM SPSS Statistics (Version 27) by the investigating dentist and a statistician (T.L.). Only the participating authors had access to the collected data.

2.9 Concern of the reviewer:

“To check the significance, the Mann-Whitney-U-test was used in the case of two independent samples and the Wilcoxon test in the case of two paired samples.” – This sentence doesn’t make sense.

Our response:

Thank you very much for this advice. We have changed the sentence as follows.

Revised Text:

To check for statistical differences, we used the Mann-Whitney-U-test for differences in the group comparisons between the case group and the control group and the Wilcoxon-test for differences between paretic and non-paretic side of one individual. 

2.10 Concern of the reviewer:

Have a section or include a section to make it distinct what the primary outcome. i.e oral health, its definition (looks like it’s a composition of several parameters) and details of how/when its measured, assuming collected at study enrolment.

Our response:

Thank you for this suggestion for improvement. We include the section 1.1 Aims of the study as part of the introduction. In this section we explained our definition of oral health and have emphasised that we realised this aim of researching oral health of patients with facial nerve palsy in a special oral health examination at the beginning of the study.

Revised Text:

1.1 Aims of the study

This study aimed to compare oral health between patients with and without FP. The endpoints to be analysed were as primary criterion the evaluation of periodontal health using the maximum periodontal screening index (PSI) as well as further various oral health parameters (see 2.3) and the oral health related quality of live (OHQoL) as secondary outcome.

Details on how and when its measured are given in the section 2.3 oral examination:

The oral examination took place in a treatment room of the Facial-Nerve-Center and took place at agreed examination appointments integrated into the daily clinic routine, without the patients performing oral hygiene beforehand. The patients sat upright on a conventional chair. The examiner, a calibrated dentist, used a blunt probe and a mirror. Each examination was performed by the same examiner. The oral cavity was illuminated with a head lamp (FaroMED-Effner; 08-365-GR; 4000-6500 LUX). The following oral health parameter were collected and documented: tooth mobility (TM)[8], periodontal screening index (PSI)[5], maximum vestibular probing depth (VPD in mm), attachment loss (AL in mm), modified sulcus bleeding index (SBI)[9], papillary bleeding index (PBI)[10], approximal plaque index (API) [11], Turesky plaque index (TI)[12], presence of tartar (POT) and dental caries by DMFT-index[13]. 

2.11 Concern of the reviewer:

Statistical analysis section needs to be expanded further, as pertinent details such as confounding, handling bias, handling missing data etc, checking model assumption are missing should be stated.

Our response:

Thank you for this comment. 

We have taken various precautions with regard to confounding and handling bias. Firstly, with regard to selection bias, the case group and the controls were matched by age and gender, which corresponds to the standardized procedure and makes both groups comparable. In addition, we reduced detection bias by conducting a standardized, objectified examination, following the same protocol for each patient. In addition, the examining dentist was calibrated before the start of the study through examination training, which limits the calibration bias. In addition, consecutive patients who appeared randomly at the clinic were examined. Thus, a random selection was made and the patients were not selected. Furthermore, to exclude bias and confounding as far as possible we included, beside the inspection of oral health, a self-management and the OHQoL and excluded specific diseases.

With regard to the aspect of handling missing data, we can say that because all patients wanted to participate there were no missing data, so we had not to calculate missing data. 

With regard to checking model assumption, in our study we only used non-parametric tests, which can be applied in all kinds of distributions and no assumptions have to be checked. 

In the manuscript, we have revised the statistical part accordingly and integrate the following text in part 2.5 Data collection and statistical analysis. 

Revised Text in statistical analysis:

In order to reduce the problem of multiple testing, we corrected our values with the Bonferroni correction.

Revised text in the discussion:

To avoid confounding and handling bias we matched patients and controls by age and gender, which makes both groups comparable. Furthermore we conducted a standardized, objectified examination, following the same protocol for each patient and calibrated the examining dentist before the start of the study through an examination training (see 2.2 Calibration of the examiner). In addition, we examined only consecutive patients who appeared randomly at the clinic. Thus, a random selection was made to avoid a selection bias. Furthermore, to exclude bias and confounding as far as possible we included, beside the inspection of oral health, a self-management and the OHQoL and excluded specific diseases. To reduce the problem of multiple testing, we corrected our values with the Bonferroni correction. Therefore we distinguished seven independent hypotheses. The individual parameters within these hypotheses are interconnected and investigated the same research question, so that not each parameter has to be corrected individually, but only the value of the respective hypothesis.

2.12 Concern of the reviewer:

Table 2 should probably be stratified by cases and controls and an overall column.

Our response:

Thank you for this comment. We removed the 2nd and 3rd column and added a total column.

Revised Text:

Table 2: Age and gender of the patients with peripheral facial nerve palsy (FPG) and the control group (CG)

Criteria FPG CG 

Total

N (%) total 43 (100.0) 43 (100.0) 86 (100.0)

Gender

N (%) male 14 (32.6) 14 (32.6) 28

 female 29 (67.4) 29 (67.4) 58

Age (in years)

Median(IQR) [Min; Max] total 51.0 (4.0) [21;73] 51.0 (5.5) [23;77] 51.0 (20.75) [21;77]

 male 50.0 (6.0) [24;73] 48.5(18.0) [24;77] 49.3(29.50) [24;77]

 female 51.0 (5.5) [21;68] 52.0 (4.0) [23;71] 51.0 (16.25) [21;71]

2.13 Concern of the reviewer:

Medians should be reported with IQRs.

Our response:

Thank you very much, we have implemented this advice.

Revised Text:

See above 2.12

2.14 Concern of the reviewer:

With so many tests was the p-value adjusted for multiple testing.

Our response:

Thank you very much for this advice. In order to reduce the problem of multiple testing, we revised our values using the Bonferroni correction. Therefore we have distinguished seven independent hypothesis. The individual parameters within the hypothesis are interrelated and investigated the same research question, so that not every parameter has to be corrected individually, but only the value of the respective hypothesis. In the Plaque infestation hypothesis, for example, there are 2 parameters, which is why we multiplied the p-value by 2 to obtain a corrected result. 

Revised Text:

Oral health parameters FPG

 CG corrected p-value 

Calculus infestation Median N(Teeth) (IQR) 

[Min; Max] 4.0 (4.0)

[0.0; 10.0] 4.0 (3.0)

[0.0; 16.0] 0.97

Plaque infestation API (%) Median (IQR) [Min; Max] 53.6* (23.7)

[14.3; 100.0] 39.3 (25.0)

[7.1; 75.0] 0.02

 TI x̅ (IQR)

[Min; Max] 0.6* (0.5)

[0.0; 1.7] 0.4 (0.3)

[0.0; 1.3] < 0.02

Gingival inflammation SBI (%) Median (IQR) [Min; Max] 58.3* (35.7)

[10.7; 100.0] 37.5 (25.9)

[3.6; 85.7] ≤ 0.002

 PBI (%) Median (IQR) [Min; Max] 26.0* (26.0)

[3.0; 62.0] 16.0 (13.0)

[0; 45.0] ≤ 0.002

Untreated caries DT x̅ (SD)

[Min; Max] 1.3* (2.0)

[0.0; 7.0] 0.5 (0.0)

[0.0; 5.0] 0.02

Missing teeth MT x̅ (SD) 

[Min; Max] 2.3 (3.7)

[0.0; 20.0] 1.9 (3.4)

[0.0; 19.0] 0.66 

Restored teeth FT Median (IQR)

[Min; Max] 10.0 (8.0)

[0.0; 20.0] 11.0 (7.0)

[0.0; 21.0] 0.46

Oral health parameters PS NPS corrected p-Value 

Calculus

infestation Median N(Teeth) (IQR) 

[Min; Max] 2.0 (2.0) 2.0 (2.0) 0.25

 [0.0; 5.0] [0.0; 5.0] 

Plaque infestation API (%) Median (IQR) 

[Min; Max] 57.1* (26.0) 46.2 (33.0) <0.02

 [14.3; 100.0] [14.3; 100.0] 

 TI x̅ (IQR)

[Min; Max] 0.6* (0.6) 0.4 (0.5) ≤0.002

 [0.0; 1.9] [0.0; 1.5] 

Gingival inflammation SBI (%) Median (IQR)

 [Min; Max] 64.3* (35.7) 57.1 (41.2) 0.06

 [7.1; 100.0] [14.3; 100.0] 

 PBI (%) Median (IQR) 

[Min; Max] 15.0* (13.0) 11.0 (16.0) ≤0.002

 [2.0; 35.0] [1.0; 30.0] 

Periodontal health VPD (mm) Median (IQR) [Min; Max] 2.4* (1.0) 2.3 (0.8) 0.08

 [0.7; 3.3] [0.6; 3.4] 

 AL (mm) Median (IQR) [Min; Max] 2.5* (1.0) 2.4 (1.0) 0.04

 [0.7; 4.3] [0.7; 4.9] 

 PSI x̅ (SD)

[Min; Max] 2.4* (0.4) 2.2 (0.5) 0.04 

 [1.5; 3.0] [1.0;3.0] 

 PSI Max x̅ (SD)

[Min; Max] 2.7* (0.5) 2.5 (0.5) 0.12

 [2.0; 3.0] [2.0;3.0] 

 TM x̅ (SD) 

[Min; Max] 0.03 (0.12) 0.03 (0.14) 1.000

 [0; 0.7] [0; 0.83] 

Untreated caries DT x̅ (IQR)

[Min; Max] 0.7 (1.2) 0.5 (0.9) 0.16

 [0.0; 4.0] [0.0; 4.0] 

Missing teeth MT x̅ (SD)

[Min; Max] 1.0 (1.9) 1.3 (2.7) 0.21 

 [0.0; 10.0] [0.0;10.0] 

Restored teeth FT Median (IQR)

[Min; Max] 6.0 (4.0) 4.0 (3.0) 0.46 

 [0.0; 11.0] [0.0; 11.0] 

2.15 Concern of the reviewer:

Table 1 can probably be written as a text.

Our response:

Thank you, that´s a good consideration. We excluded Table 1 and added in the text that one inclusion criterion was voluntary participation in the study.

Revised Text:

All consecutive included patients and healthy controls gave their written informed consent and carried out the study to the end. Edentulous, multimorbid people as well as those with diabetes mellitus type 1 and 2, hemophilia A and B, epilepsy and AIDS were excluded from the study. An existing or a past FP was considered as further exclusion criteria for the CG. CG-patients were selected if matched to FPG-patients by age and gender. 

2.16 Concern of the reviewer:

A total of 86 people aged between 21 and 77 years, 58 females (67.4%), were examined (Table 1). The median age was 51.0 years." Should this Table 2? And the sentence re median age, which group and also could benefit from giving IQRs. 

Our response:

Yes, thank you very much. There was a numbering error in the first manuscript. Now the table numbering should be correct. 

We included the IQR by age. 

Revised Text:

A total of 86 people aged between 21 and 77 years, 58 females (67.4%), were examined (Table 1). The median age for both groups was 51.0 years (IQR 20.75)."

2.17 Concern of the reviewer:

Authors should label male, female in the tables instead of signs. 

Our response:

We changed the signs to "male" and "female" in the tables.

Revised Text:

See above 2.12

3. Additional changes by the authors

3.1 References

During the revision process, we updated the reference list with some higher quality references. 

So we changed: 

Holtmann H, Hackenberg B, Wilhelm SB, Handschel J. BASICS Mund-, Kiefer- und Plastische Gesichtschirurgie [BASICS Oral and maxillofacial plastic surgery]. 2nd ed. Munich: Elsevier; 2020. pp. 104-105 to Zimmermann J, Jesse S, Kassubek J, Pinkhardt E, Ludolph AC. 2019. Differential diagnosis of Peripheral facial nerve palsy: a retrospective clinical, MRI and CSF-based study. J Neurol, 266(10):2488-2494.

Bender A, Rémi J, Feddersen B, Fesl G. Kurzlehrbuch Neurologie [Neurology textbook in brief]. 3rd ed. Munich: Elsevier; 2018. p. 328 to Santos-Lasaosa S, Pascual-Millán LF, Tejero-Juste C, Morales-Asín F.2000. Peripheral facial paralysis: etiology, diagnosis and treatment. Rev Neurol, 30(11):1048-53.

Lowden E, Baumann MA. Differentialdiagnose Pädiatrie [Differential diagnosis pediatrics]. 4th ed. In: Michalk D, Schönau E, editors. Munich: Elsevier; 2018. p. 243 to Farina R, Tomasi C, Trombelli L. 2013. The bleeding site: a multi-level analysis of associated factors. J Clin Periodontol, 40(8):735-42.

Kramer E. Prophylaxefibel Grundlagen der Zahngesundheit [Prophylaxis Primer-Basics of Dental Health]. Cologne: Deutsche Zahnärzte Verlag; 2009. p. 44 to Pitts NB, Zero DT, Marsh PD, EkstrandK, Weintraub JA,Ramos-Gomez F, Tagami J, Twetman S, TsakosG, Ismail A. 2017. Dental caries. Nat Rev Dis Primers, 3:17030.

Hellwege KD. Die Praxis der zahnmedizinischen Prophylaxe [The practice of dental prophylaxis]. Stuttgart: Thieme; 2003. pp. 30, 103 to Hague AL, Carr MP. 2007. Efficacy of an automated flossing device in different regions of the mouth. J Periodontol, 78(8):1529-37.

These changes were mainly done using study-based information deposited on Pubmed. We are convinced that this has improved the quality of the paper and hope that you agree.

And we added three references for FDI and Face:

Kahn JB, Gliklich RE, Boyev KP et al. Validation of a patient-graded instrument for facial nerve paralysis: the FaCE Scale. Laryngoscope 2001; 111:387-398.

VanSwearingen JM, Brach JS. The Facial Disability Index: reliability and validity of a disability assessment instrument for disorders of the facial neuromuscular system. Phys Ther 1996; 76: 1288-1298 Discussion 1298-1300.

Volk GF, Steigerwald F et al. Facial Disability Index und Facial Clinimetric Evaluation Skale: Validierung der Deutschen Versionen. Laryngo-Rhino-Otol 2015; 94: 163-168.

We also changed the sources of the sulcus bleeding index, the proximal plaque index and the DMFT index to more qualified sources: 

Newbrun E. Indices to measure gingival bleeding. J Periodontol 1996, 67(6):555-61.

Engelberger T, Hefti A, Kallenberger A, Rateitschak KH. Correlations among Papilla Bleeding Index, other clinical indices and histologically determined inflammation of gingival papilla. J Clin Periodontol 1983, 10(6):579-89.

Lange DE, Plagmann HC, Eenboom A, Promesberger A. [Clinical methods for the objective evaluation of oral hygiene]. Dtsch Zahnarztl Z 1977, 32(1):44-7.

3.2 CREDIT- author statements

We added the CREDIT author statements to recognise the contributions of individual authors, reduce disputes over authorship and facilitate collaboration in the following way:

Author Contributions: conceptualization: L.S., G.F.V., I.M.S; methodology: L.S., G.F.V., I.M.S., A.-M.K.; software: L.S.; validation: L.S., G.F.V., C.B., I.M.S; formal analysis: L.S., T.L.; investigation: L.S.; resources: G.F.V., T.L., I.M.S; data curation: L.S.; writing—original draft preparation: L.S.; writing—review and editing: L.S., G.F.V., A.-M.K., C.B., T.L., I.M.S.; visualization: L.S., supervision: G.F.V., I.M.S.; project administration: G.F.V., I.M.S.; funding acquisition: no funding

3.3 Preventions suggestions and advices for patients

During the extensive and collaborative review process, we thought about ways to improve the oral health of patients with facial nerve palsy and established advice for affected patients. We would now like to integrate our thoughts into the paper, so that in the future the oral problems of the patients can be addressed more specifically in the therapy process. For this purpose, we also developed a flyer, which we would like to make available as additional material.

---

## [Decision Letter · Decision Letter 1]

20 Jul 2022

PONE-D-21-21348R1Oral health and oral health-related quality of life in patients with chronic peripheral facial nerve palsy with synkineses- a case-control-study

PLOS ONE

Dear Dr. Strobelt,

Thank you for submitting your manuscript to PLOS ONE. After careful consideration, we feel that it has merit but does not fully meet PLOS ONE’s publication criteria as it currently stands. Therefore, we invite you to submit a revised version of the manuscript that addresses the points raised during the review process.

We look forward to receiving your revised manuscript.

Kind regards,

Andrej M Kielbassa, Prof. Dr. med. dent. Dr. h. c.

Academic Editor

PLOS ONE

Reviewers' comments:

Reviewer's Responses to Questions

**Comments to the Author**

1. If the authors have adequately addressed your comments raised in a previous round of review and you feel that this manuscript is now acceptable for publication, you may indicate that here to bypass the “Comments to the Author” section, enter your conflict of interest statement in the “Confidential to Editor” section, and submit your "Accept" recommendation.

Reviewer #1: All comments have been addressed

Reviewer #2: (No Response)

Reviewer #3: All comments have been addressed

2. Is the manuscript technically sound, and do the data support the conclusions?

Reviewer #1: Yes

Reviewer #2: Partly

Reviewer #3: Yes

3. Has the statistical analysis been performed appropriately and rigorously? 

Reviewer #1: Yes

Reviewer #2: Yes

Reviewer #3: Yes

4. Have the authors made all data underlying the findings in their manuscript fully available?

Reviewer #1: No

Reviewer #2: No

Reviewer #3: Yes

5. Is the manuscript presented in an intelligible fashion and written in standard English?

Reviewer #1: Yes

Reviewer #2: No

Reviewer #3: Yes

6. Review Comments to the Author

Reviewer #1: (No Response)

Reviewer #2: ------------------------------------------------------------------------------------------------------

See attached file.

Reviewer #3: This is an interdisciplinary study including patients with facial nerve palsy (FP) and a control group. It was shown that oral health parameters in patients with FP were significantly worse than in the control group. Several other variables related to the disease were shown to be significantly affected as well.

This is a revised version of a manuscript with an important topic. While reading it, it seems that the authors have responded in a satisfactory way to the reviewer´s comments. This led to an improvement of the manuscript and as such I suggest accepting the manuscript.

7. PLOS authors have the option to publish the peer review history of their article (what does this mean?). If published, this will include your full peer review and any attached files.

Reviewer #1: No

Reviewer #2: No

Reviewer #3: No

---

## [Author Response · Author response to Decision Letter 1]

31 Aug 2022

to Reviewer 1/Prof. Dr. med. dent. Dr. h. c. Andrej M Kielbassa: Thank you very much for your assessment of our manuscript. In the document "Response to Reviewers" we have addressed your helpful comments point by point and have made the appropriate changes to the manuscript. We very much hope that our correction meets your expectations. 

to Reviewer 3: Thank you very much for your assessment of our revised manuscript. We are very pleased that the corrections made have been implemented satisfactorily for you and that you recommend acceptance of our manuscript.

---

## [Decision Letter · Decision Letter 2]

3 Oct 2022

Oral health and oral health-related quality of life in patients with chronic peripheral facial nerve palsy with synkineses- a case-control-study

PONE-D-21-21348R2

Dear Dr. Strobelt,

we are pleased to inform you that your manuscript has been judged scientifically suitable for publication and will be formally accepted for publication once it meets all outstanding technical requirements. Congratulations, and stay healthy!

Kind regards,

Andrej M Kielbassa, Prof. Dr. med. dent. Dr. h. c.

Kind regards,

Andrej M Kielbassa

Academic Editor

PLOS ONE

Reviewers' comments:

Reviewer's Responses to Questions

**Comments to the Author**

1. If the authors have adequately addressed your comments raised in a previous round of review and you feel that this manuscript is now acceptable for publication, you may indicate that here to bypass the “Comments to the Author” section, enter your conflict of interest statement in the “Confidential to Editor” section, and submit your "Accept" recommendation.

Reviewer #1: All comments have been addressed

Reviewer #2: All comments have been addressed

Reviewer #3: All comments have been addressed

2. Is the manuscript technically sound, and do the data support the conclusions?

Reviewer #1: Yes

Reviewer #2: Yes

Reviewer #3: Yes

3. Has the statistical analysis been performed appropriately and rigorously? 

Reviewer #1: Yes

Reviewer #2: Yes

Reviewer #3: Yes

4. Have the authors made all data underlying the findings in their manuscript fully available?

Reviewer #1: No

Reviewer #2: Yes

Reviewer #3: Yes

5. Is the manuscript presented in an intelligible fashion and written in standard English?

Reviewer #1: Yes

Reviewer #2: Yes

Reviewer #3: (No Response)

6. Review Comments to the Author

Reviewer #1: (No Response)

Reviewer #2: With the help of the reviewers, the Co-Authors have considerably improved their draft. This revised manuscript is considered ready for external review.

Reviewer #3: The authors have adequately addressed all the comments raised in the previous rounds of review.

Due to the importance and novelty of the data I recommend to accept the manuscript for publication.

7. PLOS authors have the option to publish the peer review history of their article (what does this mean?). If published, this will include your full peer review and any attached files.

Reviewer #1: No

Reviewer #2: No

Reviewer #3: No
